# Changes in Ovarian Activity and Expressions of Follicle Development Regulation Factors During the Laying–Incubation Cycle in Magang Geese

**DOI:** 10.3390/ani15101390

**Published:** 2025-05-12

**Authors:** Rui Wu, Junfeng Sun, Jianqiu Pan, Xu Shen, Danli Jiang, Hongjia Ouyang, Danning Xu, Yunbo Tian, Yunmao Huang

**Affiliations:** College of Animal Science and Technology, Zhongkai University of Agriculture and Engineering, Guangzhou 510225, China; wurui779@outlook.com (R.W.); 17806709411@163.com (J.S.); panjianqiuzhku@163.com (J.P.); shenxuxu3@gmail.com (X.S.); danli0222@163.com (D.J.); oyhj@zhku.edu.cn (H.O.); xdanning@126.com (D.X.); tyunbo@126.com (Y.T.)

**Keywords:** laying–incubation cycle, ovary, follicle, regulation, Magang geese

## Abstract

Compared to chicken and duck production, goose industry development has been significantly delayed, primarily due to their seasonal reproductive behavior and strong broody tendencies. In this study, ovarian morphology, follicular development, blood reproductive hormones, and the expressions of reproductive regulators in the gonadal stroma, follicular granulosa, and theca layer of the follicles during the laying–incubation cycle in Magang geese (*Anser domesticus*) were examined. Our results revealed that during the breeding season of Magang geese, the hormone receptors in the ovaries showed corresponding changes that jointly regulated the steroidogenesis pathway and follicular development. These changes occurred alongside fluctuations in upstream hormones throughout the laying–incubation cycle, reflecting the cyclic changes in reproductive hormones upstream from the reproductive axis, with changes in ovarian activity and follicle development.

## 1. Introduction

Domestic geese (*Anser domesticus*) are prominent among the earliest domesticated animals. For centuries, they have been a vital source of diverse consumer products, including succulent meat, luxurious foie gras, and warm down feathers, in numerous Asian and European countries [1]. Despite their long-standing significance, the goose industry has persistently trailed behind other poultry sectors, such as chickens and ducks. This developmental disparity is predominantly attributed to the constraints imposed by the reproductive activities and performance of geese. Geese exhibit a notably lower egg-laying performance than other poultry species, a characteristic intricately linked to their unique features in follicular development, ovulation, and egg-laying regulation. These regulatory mechanisms differ significantly from those of well-studied poultry like chickens and ducks, manifesting primarily at three hierarchical levels: the seasonal (annual) cycle, the laying–incubation cycle, and the egg production cycle [1]. 

Magang geese, a typical short-day seasonal breeding avian species, are distinguished by their distinct seasonality and pronounced broodiness [2]. In contrast to many long-day breeding goose species, which either lack brooding behavior during the laying season or only exhibit it in the latter stages of their annual reproductive cycle, Magang geese are prone to experiencing multiple brooding episodes. Within a single reproductive season, they may undergo four to five distinct laying–incubation cycles [3]. This recurrent broodiness has become a major bottleneck, severely limiting their reproductive efficiency.

It is widely established that avian reproductive activity is primarily regulated by the hypothalamic–pituitary–gonadal (HPG) axis [4,5]. The hypothalamus, acting as the central integration and regulatory hub for avian reproduction, secretes gonadotropin-releasing hormone (GnRH) and gonadotropin-inhibitory hormone (GnIH). These hormones modulate the synthesis and release of pituitary gonadotropins [6,7,8]. Subsequently, luteinizing hormone (LH) and follicle-stimulating hormone (FSH) act on the gonads, promoting gametogenesis and sex steroid hormone synthesis, thereby influencing gonadal development and ovulation [9]. Prolactin (PRL), in addition to the aforementioned regulatory mechanisms, has been identified as a key inhibitory hormone in avian reproduction [10]. Abundant studies have demonstrated that persistently elevated circulating PRL levels suppress hypothalamic GnRH expression. This suppression leads to reduced pituitary gonadotropin secretion and the down-regulation of steroidogenic enzymes, ultimately disrupting follicular development and ovulation, causing ovarian regression and inducing broodiness in avian species [11].

Under the precise regulation of the central nervous system, the HPG axis forms a comprehensive feedback mechanism. This system enables coordinated yet mutually restrictive interactions among reproduction-related factors, maintaining dynamic equilibrium in reproductive hormones and achieving the precise regulation of avian reproductive activities. Importantly, the biological functions of various reproductive hormones are realized through their interaction with specific receptors in target cells. Therefore, we investigated the changes in ovarian activity, reproductive hormones, and expressions of follicle development regulation factor in gonads during the laying–incubation cycle in Magang geese’s breeding season to reveal the important regulatory role of reproductive hormones and their receptors on follicular development and ovarian activity in gonads. 

## 2. Materials and Methods

### 2.1. Animal Experimental Design

A total of 1200 2-year-old female healthy Magang geese (♂:♀ = 1:5) in the middle of the breeding period were selected to observe their nesting behavior (in natural light). In practical production, brooding geese were kept in a different location for 10 days with restricted feed to forcibly terminate their broodiness, and then these geese were returned to the flock of geese. The ovarian morphology and follicular development of the geese were observed during the laying period (laying stage), on the 2nd day (early broodiness, ErB stage), 5th (the depth of broodiness, DpB stage) day after brooding, and 7th (the end of broodiness, EdB stage) day when the forced termination of broodiness took place, and the types of follicles were counted and the gonadosomatic index (GSI) was calculated (Weight/gonadal weight × 100%). Blood samples (*n* = 6) were collected from the wing vein, and the serum was centrifuged at 4000 rpm for 10 min and then stored at −20 °C for hormone analysis. Tissue samples from the ovaries (after the removal of surface follicles), small white follicles (<6 mm, SWFs), large white follicles (6~8 mm, LWFs), and the granulosa and theca layers of the small yellow follicles (8~10 mm, SYFs) and large yellow (grade) follicles (>10 mm, LYFs) were collected and stored in liquid nitrogen until gene and protein expression analysis (*n* = 6).

### 2.2. Blood Hormone Detection

Blood follicle-stimulating hormone (FSH, *EK10443*), luteinizing hormone (LH, *EK11984*), prolactin (PRL, *EK8232*), progesterone (P4, *EK8164*) and estradiol (E2, *EK20412*) levels are measured using ELISA kits (Signalway Antibody, Maryland, CO, USA), following the instructions for each product. The blood inhibin (INH, *MBS260380*) level is measured using ELISA kits (MyBiosource, SanDiego, CA, USA), following the instructions for the product.

### 2.3. Gene Expression Detection

Total RNA was isolated from the ovaries and follicles of female geese with Trizol (Foster City, CA, USA), and the concentration was measured by absorbance at 260 nm using a spectrophotometer (a 260/280 ratio > 1.8). The complementary DNA (cDNA) was synthesized using a PrimeScriptTMRT reagent kit with gDNA eraser (Kusatsu, Shiga-Ken, Japan). The primers for the real-time quantitative polymerase chain reaction (RT-qPCR) were designed according to the mRNA sequence of goose in GenBank (Table 1) and then synthesized by Sangon Biotech Co., Ltd. (Shanghai, China). RT-qPCR was performed using a QuantStusio 7 Flex (Foster City, CA, USA) with a pre-denaturation step at 50 °C for 2 min and then at 95 °C for 10 min for one cycle and 95 °C for 15 s and then the annealing temperature for 1 min for 40 cycles. Reactions were performed in a final volume of 10 μL using SYBR Green PCR Master Mix (Nanjing, China) and 2.5 pmol primers. This process for each sample was repeated 3 times. The results were analyzed using the 2^−ΔΔCT^ method. β-actin was used as the reference gene [12].

### 2.4. Protein Expression Detection

The protein from the ovaries and follicles were separated by electrophoresis, transferred to the polyvinylidene difluoride (PVDF) membrane at 4 °C, and then blocked with 5% skimmed milk for 2 h at room temperature, and the membranes were incubated with the primary antibody overnight at 4 °C. The primary antibodies used were rabbit polyclonal FSHR (HuBei, China), LHR (JiangSu, China), StAR (JiangSu, China), 3β-HSD (JiangSu, China), CYP19A1 (Shandong, China), GnIH (self-made-polyclonal antibody) and mouse monoclonal β-actin (JiangSu, China). The membranes were washed 3 times using 1× phosphate-buffered saline with 0.05% Tween 20 (PBST) each for 10 min. The membranes were then incubated with peroxidase-conjugated secondary antibodies at room temperature for 1 h and washed 3 times at PBST each for 5 min. The membranes were incubated with an ECL kit (Claremont, CA, USA) and visualized using Tanon (Shanghai, China). The protein bands were analyzed in grayscale using the Image J software (Image Pro Plus 6.0).

### 2.5. Statistical Analysis

All values were expressed as the mean ± SEM. The vitro experimental data of RT-qPCR and the Western blot data were analyzed by a one-way ANOVA using the IBM SPSS Statistics 26 (IBM SPSS software, Inc., Chicago, IL, USA) and Tukey’s test for the significant difference analysis. Graphs were plotted using GraphPad Prism 8.0 software (Prism Software, Inc., San Diego, CA, USA). *p* < 0.05 was considered statistically significant.

## 3. Results

### 3.1. Ovarian Morphology and Follicular Development in the Laying–Incubation Cycle

The results for ovarian morphology and follicular development in various stages of the laying–incubation cycle in Magang geese showed that the follicles were fully developed in the laying stage, with an amount of LYFs, SYFs and LWFs and a GSI value that were significantly higher than those in the ErB, DpB and EdB stages (Figure 1 and Figure 2). Ovaries began to atrophy in the ErB stage, with no LYFs and a small amount of SYFs and LWFs. With increasing atrophy, the number of SYFs growing in the ovaries decreased in the DpB stage. Ovarian development resumed and the number of LWFs increased in the EdB stage.

### 3.2. Reproductive Hormone Changes in the Laying–Incubation Cycle

The plasma levels of reproductive hormones are shown in Figure 3. The results showed that FSHs were significantly higher in the laying stage than at the EdB stage, P4 was significantly higher in the laying stage than in the other three stages, E2 was significantly higher in the laying stage than at the ErB stage, PRL was significantly lower in the laying stage than at the ErB stage, and INH was significantly higher in the laying stage than that in the ErB and EdB stages, and there was no significant difference in plasma LH concentrations among the four reproduction stages.

### 3.3. Expression Levels of Reproductive Factors in Ovarian Stroma in the Laying–Incubation Cycle

The expression levels of reproduction factors showed that the change trend of *LHR* was the same as that of *FSHR* and opposite to that of *GnIHR* (Figure 4A–C), but there were no significant differences. The relative expression of *GnIH* in the ErB stage was significantly higher than in the DpB stage (Figure 4D). The relative expression of *StAR* was significantly higher in the ErB stage than in the laying stage (Figure 4F), and *3β-HSD* and *CYP19A1* presented the same change trends (Figure 4G,H). There were no significant differences in the relative expression levels of *PRLR*, *INHA* and *INHB* (Figure 4E,I,J).

The protein expression level of CYP19A1 in ovarian stroma in the laying stage was significantly higher than in the EdB stage. There were no significant differences in the protein expression levels of FSHR, LHR, and StAR in the four stages (Figure 5).

### 3.4. Expression Levels of Reproductive Factors at Different Stages of Follicular Development

The expression levels of reproduction factors showed that *FSHR* and *GnIH*/*GnIHR* were mainly expressed in the granulosa layer of large yellow follicles, and the expression levels gradually decreased as the follicles developed into preovulation follicles, and they were significantly higher in the F6 granulosa layer (F6G) than in the F1 granulosa layer (F1G) (Figure 6A,C,E). *LHR* was highly expressed in both the granulosa and membrane layers of large yellow follicles, and its expression gradually increased with follicle development, reaching its highest level before ovulation, and the relative expression level of *LHR* in the F6G was significantly lower than in the F1G (Figure 6B). The expression level of *PRLR* was relatively higher in the early stage of follicular development, with the highest expression level in the membrane layer of small yellow follicles and large white follicles, and its relative expression in the SY theca layer (SYT) was the highest (Figure 6D). *StAR* and *3β-HSD* were expressed in both the granulosa and membrane layers, and the expression levels gradually increased with follicular development. The relative expression level of *StAR* was the highest in the F1G (Figure 6F), and the relative expression level of *3β-HSD* in the F3 granulosa layer (F3G) was significantly higher than that of other developmental-stage follicles (Figure 6G). *CYP19A1* was mainly expressed in the membrane layer, gradually increasing during the early stage of follicular development and gradually decreasing after entering the hierarchical development stage. The expression level was also higher in the early development stage and the lowest in the F1 theca layer (F1T) before ovulation in the granulosa layer (Figure 6H).

The protein expression results showed that the protein expression levels of FHSR, LHR, and PRLR were basically consistent with the changes in gene expression levels. The protein expression of the three was mainly present in the granulosa layer and gradually increased with the development of follicles and decreased before ovulation. The highest protein level of FSHR, LHR and PRLR was in the granulosa layer of SYG or F3G (Figure 7A–C). There was no significant difference in the expression of GnIH between the granulosa and membrane layers, and the expression levels decreased with the development of follicles. The expression level of GnIH in the granulosa layer increased again before ovulation, but there were no significant differences in the protein expression levels at different follicle development stages (Figure 7D).

## 4. Discussion

In this study, we obtained a primary understanding of the morphological characteristics of the ovaries and follicles during the laying–incubation cycle in Magang geese. With the onset of brooding, ovarian activity degenerated and atrophied. The number of LWFs and SYFs decreased rapidly, and LFYs disappeared. With the termination of brooding, ovarian activity was gradually restored. The number of LWFs and SYFs increased gradually, and LFYs began to appear. These morphological changes show alignment with those seen in Zhedong white geese [13].

The HPG axis mainly regulates avian reproductive activity. FSH and LH, which are released by the pituitary gland, are the two main regulatory factors that promote ovarian follicular development and ovulation [14,15], whereas the enhanced secretion of PRL decreases the secretion of FSH and LH to inhibit follicular development and cause the occurrence of broodiness [16]. In this experiment, the high plasma levels of FSH and LH stimulated gonadal development in the laying stage, and the synthesis and secretion of P4, E2, and INH were improved with follicular development. With continued laying, PRL secretion increased, FSH and LH secretion decreased, the ovary was degraded, and P4, E2, and INH were significantly decreased. These are consistent with the research results on chickens [17]. These reproductive hormones all had a clear pattern of change in the laying–incubation cycle.

The gonadotrophins secreted by the pituitary gland need to bind to corresponding receptors to realize the function of promoting gonadal development [18]. In this experiment, we found that the expression of *FSHR* and *LHR* in the ovarian stroma was high during the laying period, decreased after the occurrence of incubation, and increased with the forced termination of broodiness, although the differences were not significant. The expression levels of *GnIH*/*GnIHR* showed an opposite trend, with low levels during the laying period and high levels after incubation, and then they rose again with the termination of broodiness. *PRLR* gradually increased with the formation of incubation and began to decrease with broodiness termination, but the difference was insignificant. The insignificant differences in these receptors might be related to their expression mainly in the granulosa and membrane layers of follicles [19]. The synthesis pathways of steroids, including *StAR*, *3β-HSD*, and *CYP19A1*, gradually increased with the occurrence of incubation and then returned to normal with the termination of broodiness. This indicated that the steroid pathway strengthens with continued laying, leading to increased progesterone secretion, elevated PRLR expression, and decreased FSHR and LHR expression in the gonad. At the same time, PRL secretion increased, and *GnIH*/*GnIHR* in the gonad also increased, ultimately leading to the formation of an incubation period. These results indicated that the expression of FSHR, LHR, GnIHR and PRLR in the gonads was consistent with changes in gonadal activity and corresponded to the upstream secretion of reproductive hormones, jointly regulating gonadal activity [20]. The secretion of inhibin showed no significant changes, but the change trend was consistent with the secretion of INH. The protein expression changes in FSHR, LHR, and the steroidogenic factors StAR and CYP19A1 showed a consistent trend, gradually increasing with continued laying and incubation occurring and decreasing with broodiness and the termination of incubation. As an important factor upstream of the reproductive axis, GnIH could be directly expressed and regulate the expression of FSHR and LHR at the gonad level [21]. What important regulatory roles do the gonadal expression of GnIH and GnIHR play in the expression of these receptors? Which upstream factors regulate the expression of gonadal GnIH/GnIHR? These questions require further in-depth research.

In addition to detecting the expression of the main regulatory factors of follicular development and steroid pathway factors in the ovarian stroma, we focused on detecting the expression of the main regulatory factors in the granulosa and membrane layers of follicles at different developmental stages, which are the main secretion sites of P4 and E2 [22,23,24]. The results showed that *FSHR* and *GnIH*/*GnIHR* were mainly secreted in the granulosa layer, gradually increasing during the early follicular development stage, reaching the highest level after the hierarchical development stage, and decreasing before ovulation. The secretion of *FSHR* and *GnIH*/*GnIHR* showed a similar change trend. This indicated that FSHR and GnIH/GnIHR dynamically maintained follicular development from both promoting and inhibiting aspects, and their secretion levels were positively correlated with the speed of follicular development. The above results were consistent with the research findings that FSHR acts mainly in the granulosa layer of the follicle and plays an important role in follicular selection [25,26]. *LHR* was expressed in both the granulosa and membrane layers, mainly in the late stage of follicular development, and its expression level increased rapidly with the rapid development of follicles. This was consistent with the fact that LH mainly cooperates with FSH to promote the rapid development of follicles in the late stage and promote the final stage of ovulation [27]. The above results fully demonstrated that the regulation of corresponding receptor expression in the ovary was crucial to regulate follicular development, in addition to the upstream hormone of the reproductive axis, and both jointly regulated follicular development and brooding.

Numerous studies have demonstrated that GnIH primarily regulates the secretion of pituitary gonadotropins, which are upstream components of the reproductive axis. This regulation occurs through direct or indirect mechanisms, ultimately modulating follicular development [28,29]. Moreover, studies have shown that GnIH can regulate the expression of FSHR and LHR by interacting with GnIHR in the ovary [30]. The results of this study confirmed that the ovaries can directly express GnIH and GnIHR, indicating that the GnIH binding to GnIHR in the ovary did not come from upstream of the reproductive axis but directly from the gonad. PRLR has a high expression level in the membrane layer during the early stage of follicular development, a low expression level in the pre-follicular granulosa layer, and a stable expression level in the membrane layer and granulosa layer during other stages of follicular development. This may be related to the research results that high-level PRL inhibits follicular development, moderate concentrations of PRL are necessary for follicular development, and low-level PRL may affect follicular development [31,32]. However, more research is needed to reveal these specific relationships. Additionally, we observed an insignificant protein expression of various components in F6G during the experiments. The small size of F6 follicles may have led to yolk contamination during granulosa layer isolation, potentially affecting the results. Therefore, protein expression data from this tissue were excluded from further analysis to prevent confounding effects. In subsequent experiments, a comparable phenomenon was observed when detecting small-molecule proteins (GnIH) in F3G and SYG. This suggests that yolk contamination may interfere with protein expression measurements, particularly for low-molecular-weight proteins. Therefore, the protein expression results obtained from follicles at different developmental stages in this study should be interpreted cautiously and presented solely as supplementary reference data without an in-depth discussion.

Studies have shown that FSH and LH could regulate follicular development by activating the p38 MAPK and TGF-β signaling pathways to affect the steroidogenesis pathway [33,34,35]. GnIH could affect the expression of steroid pathway factors, such as StAR, 3β-HSD, and CYP19A1, to inhibit the synthesis of P4 and E2 [36]. In this experiment, the expressions of *StAR*, *3β-HSD* and *CYP19A1* were low in the early follicular stage and gradually increased with rapid follicular development. The *StAR* expression level reached the highest level before ovulation, while the expressions of *3β-HSD* and *CYP19A1* decreased before ovulation. *StAR* and *3β-HSD* mainly changed in the granulosa layer, while *CYP19A1* was expressed primarily in the membrane layer. These results were consistent with studies on chickens [37]. There was no significant difference in the expression levels of FSHR, LHR, PRLR and GnIH during the different follicular development stages. However, the changing trends in FSHR, LHR, and PRLR protein levels were consistent with the changes in gene expressions, increasing with follicular development and decreasing before ovulation.

Many studies have shown that GnIH could directly inhibit the secretion of pituitary gonadotropins or indirectly affect the secretion of pituitary gonadotropins by regulating the secretion of GnRH in the hypothalamus to inhibit avian reproduction [38,39,40]. There are many research reports on the regulation mechanism of GnIH on avian reproduction, but most of them have focused on the regulation at hypothalamic–pituitary level [41,42]. Many studies have shown that GnIH could directly participate in the regulation of follicular development at the gonadal level [43]. Our study confirmed the gonadal expression of GnIH, indicating its potential regulatory role in seasonal transitions of reproductive activities and the laying–incubation cycle at the gonadal level. GnIH may play a crucial role in follicular development during the breeding season, cooperatively regulating reproductive behavior with FSHR.

These findings indicate that GnIH/GnIHR interactions function throughout all stages of the HPG axis. Comprehensive future research integrating both upstream and downstream components of the reproductive axis would elucidate GnIH’s mechanisms more thoroughly, potentially leading to GnIH-based vaccine development as an innovative approach for improving production efficiency. Notably, we consistently detected LWF presence across various reproductive phases. A further investigation is needed to determine whether LWFs exhibit similar patterns of reproductive factor variation during different reproductive stages.

## 5. Conclusions

In conclusion, our results revealed that FSHR, LHR and PRLR in the ovary presented corresponding changes to jointly regulate the steroidogenesis pathway and follicular development with upstream hormones in the laying–incubation cycle during the breeding season of Magang geese, in addition to the cyclic changes in reproductive hormones upstream of the reproductive axis, such as FSH, LH, and PRL, with changes in ovarian activity and follicle development. The expression levels of FSHR, LHR, and PRLR showed either promoting or inhibiting changes at different stages of follicular development. GnIH/GnIHR was directly expressed in the ovary and its follicles, and the expression of GnIH/GnIHR showed consistent changes with FSHR at different stages of follicular development. This indicated that GnIH and FSH might dynamically regulate follicular development from both positive and negative aspects, and GnIH/GnIHR might have a regulatory effect on the expressions of FSHR, LHR and PRLR in ovaries, which had been confirmed in our other experiments [30,36].

## Figures and Tables

**Figure 1 animals-15-01390-f001:**
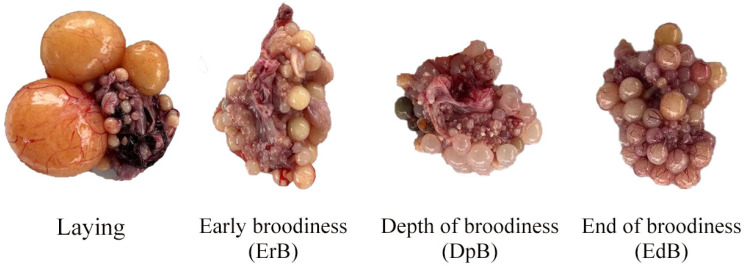
Ovarian morphology at different stages of the laying–incubation cycle.

**Figure 2 animals-15-01390-f002:**
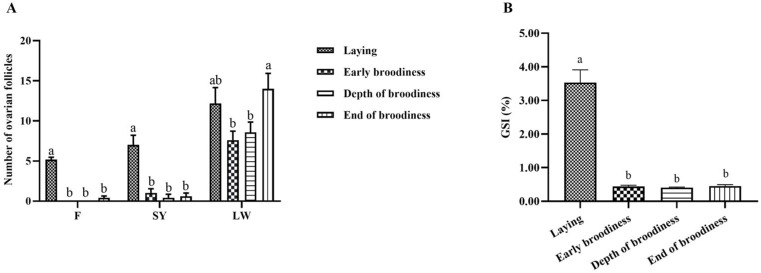
The types and number of follicles (**A**) and the GSI (**B**) at different stages of the laying–incubation cycle. Significant differences are indicated by different lowercase letters (such as a, b) (*p* < 0.05).

**Figure 3 animals-15-01390-f003:**
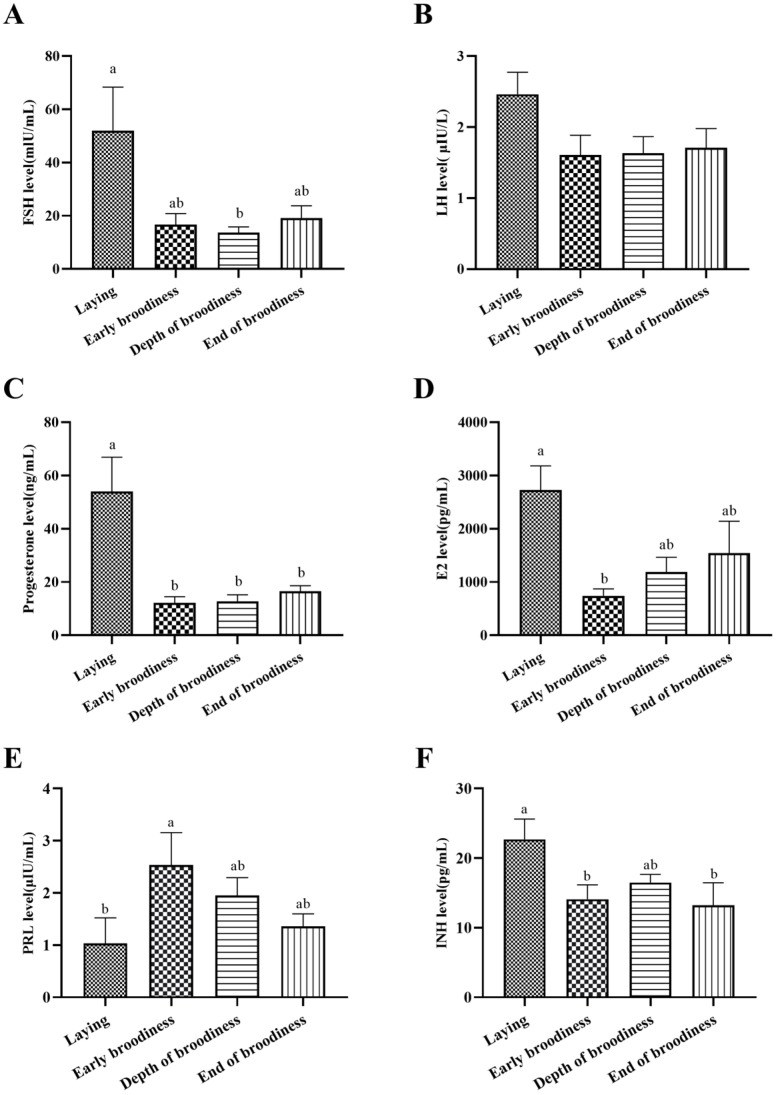
Changes in blood hormones in the laying–incubation cycle. (**A**) The plasma hormone levels of FSH; (**B**) the plasma hormone level of LH; (**C**) the plasma hormone level of P4; (**D**) the plasma hormone level of E2; (**E**) the plasma hormone level of PRL; (**F**) the plasma hormone level of INH. Significant differences are indicated by different lowercase letters (such as a, b) (*p* < 0.05).

**Figure 4 animals-15-01390-f004:**
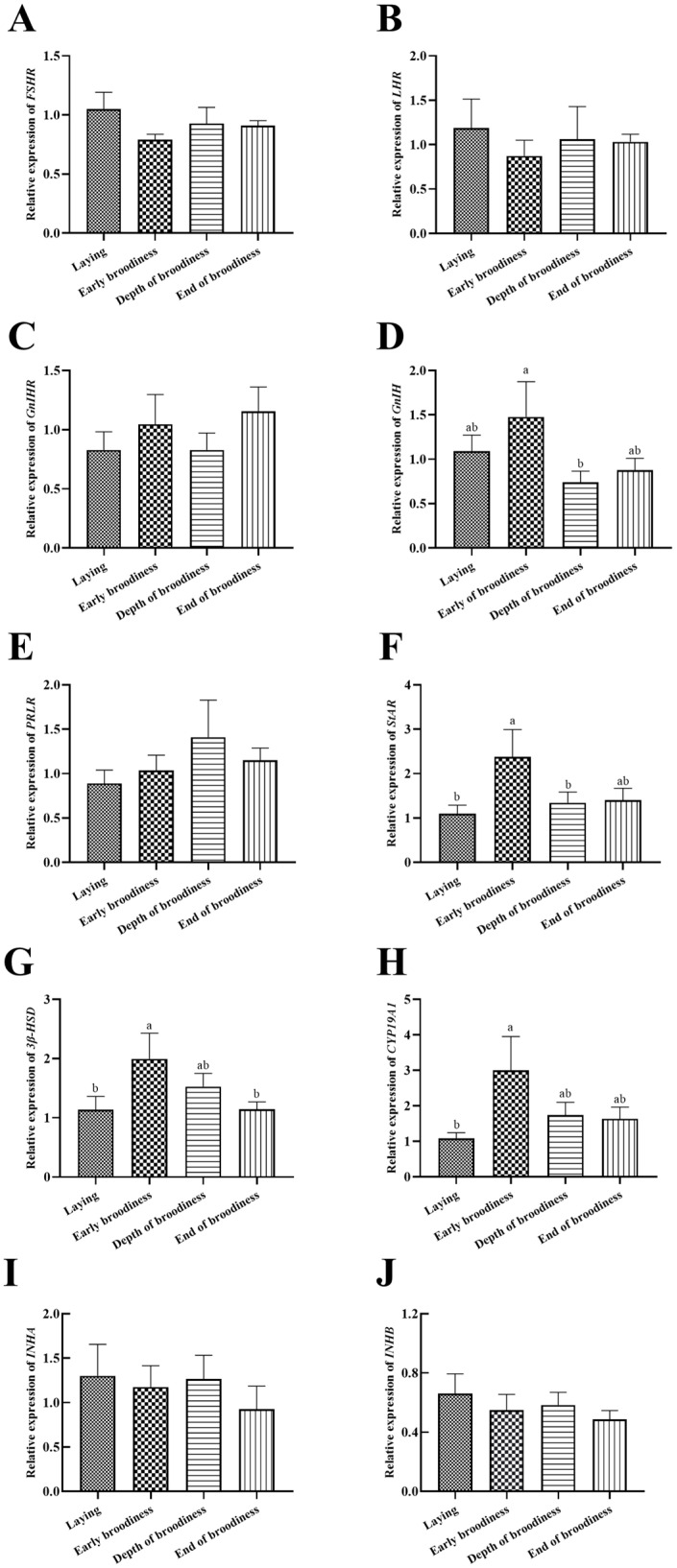
Gene expression levels of reproductive factors in ovarian stroma in the laying–incubation cycle. (**A**) Relative expression level of *FSHR*; (**B**) relative expression level of *LHR*; (**C**) relative expression level of *GnIHR*; (**D**) relative expression level of *GnIH*; (**E**) relative expression level of *PRLR*; (**F**) relative expression level of *StAR*; (**G**) relative expression level of *3β-HSD*; (**H**) relative expression level of *CYP19A1*; (**I**) relative expression level of *INHA*; (**J**) relative expression level of *INHB*. Significant differences are indicated by different lowercase letters (such as a, b) (*p* < 0.05).

**Figure 5 animals-15-01390-f005:**
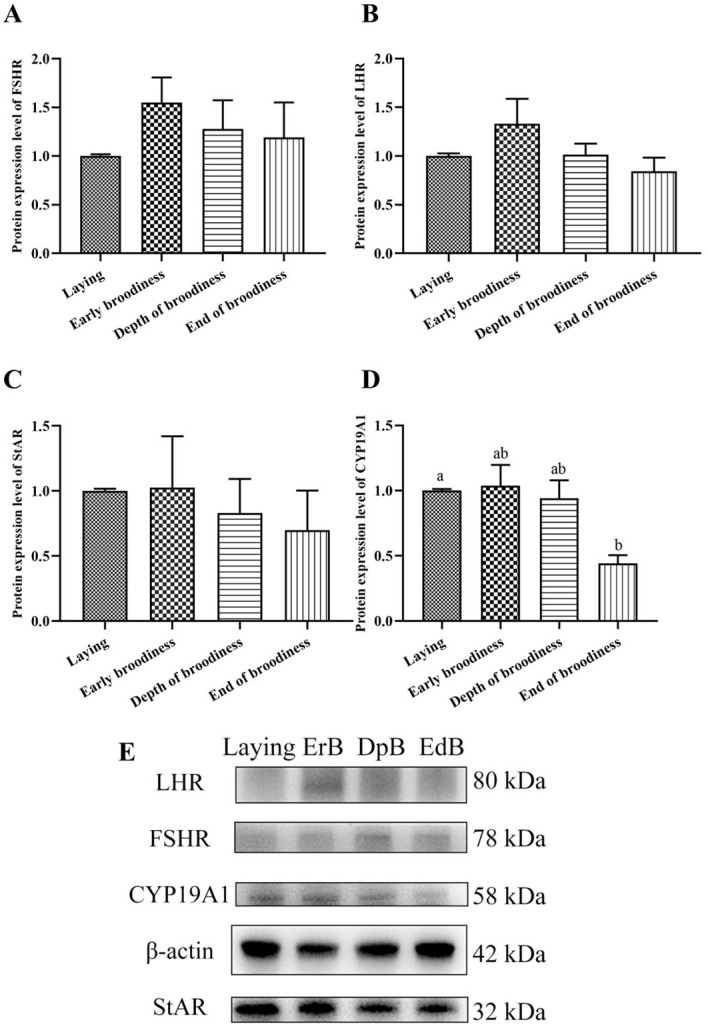
Protein expression levels of reproductive factors in ovarian stroma in the laying–incubation cycle. (**A**) The protein expression level of FSHR; (**B**) the protein expression level of LHR; (**C**) the protein expression level of StAR; (**D**) the protein expression level of CYP19A1; (**E**) Western blots of FSHR, LHR, StAR and CYP19A1. Significant differences are indicated by different lowercase letters (such as a, b) (*p* < 0.05).

**Figure 6 animals-15-01390-f006:**
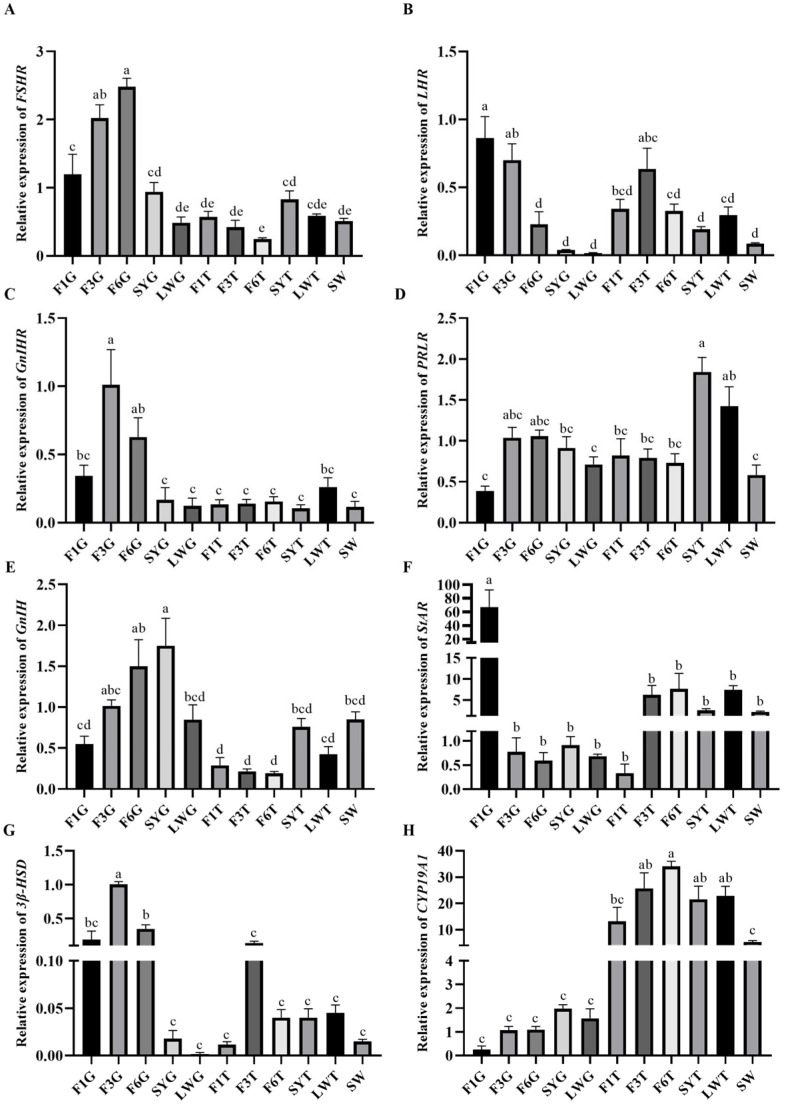
Gene expression levels of reproductive factors at different stages of follicular development. (**A**) The relative expression level of *PRLR*; (**B**) the relative expression level of *LHR*; (**C**) the relative expression level of *FSHR*; (**D**) the relative expression level of *GnIHR*; (**E**) the relative expression level of *StAR*; (**F**) the relative expression level of *3β-HSD*; (**G**) the relative expression level of *CYP19A1*; (**H**) the relative expression level of *GnIH* in follicles. Follicles were collected during the laying stage. The LYFs are categorized as F1, F2, F3, etc., from large to small. The granulosa layer and theca layer isolated from the same follicle were labeled as G and T, respectively—for example, F1G for the granulosa layer of F1 and F1T for the theca layer of F1. Significant differences are indicated by different lowercase letters (such as a, b, c, d, e) (*p* < 0.05).

**Figure 7 animals-15-01390-f007:**
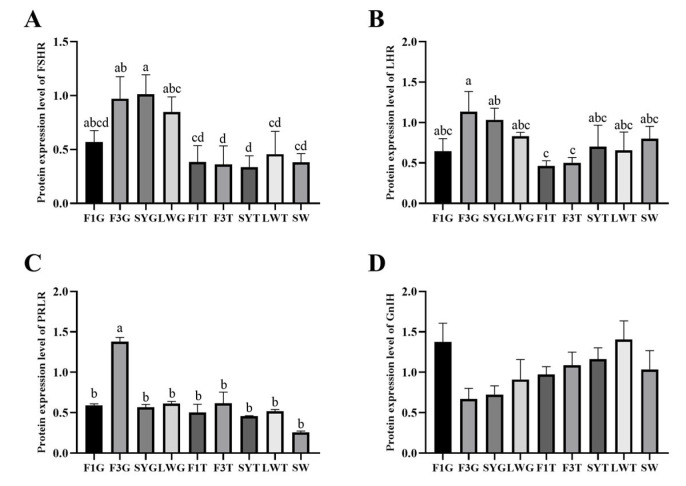
Protein expression levels of reproductive factors at different stages of follicular development. (**A**) The protein expression level of FSHR; (**B**) the protein expression level of LHR; (**C**) the protein expression level of PRLR; (**D**) the protein expression level of GnIH; (**E**) Western blots of FSHR, LHR, PRLR and GnIH. Significant differences are indicated by different lowercase letters (such as a, b, c, d) (*p* < 0.05).

**Table 1 animals-15-01390-t001:** Primers for real-time quantitative PCR.

Gene	Primer Sequence (5′-3′)	Tm (°C)	PCR Product (bp)
*FSHR*	F: GATGAGCAACCTGGCAATAAG	58	140
	R: GGTGAGCAAGCCACATTAAC		
*LHR*	F: GTAACACTGGAATAAGGGAAT	57	191
	R: GAAGGCATGACTGTGGATA		
*PRLR*	F: ACGAGTTGCGACTAAAGCCT	60	228
	R: CACCCACGATGATCCACACA		
*GnIHR*	F: CATCCTGGTGTGCTTCATCG	56	164
	R: ACATGGTGTTGTCAAAGGGC		
*StAR*	F: CTGCCATCTCCTACCGCCAC	60	217
	R: CTGCTCCACCACCACCTCCA		
*3β-HSD*	F: AGAAGTGACAGGCCCAAACT	60	188
	R: ACATGGATCTCAGGGCACAA		
*CYP19A1*	F: GGATGGGAGTAGGTAATGCC	60	274
	R: ACAAGACCAGGACCAGACAG		
*GnIH*	F: AAAGTGCCAAATTCAGTTGCT	58	120
	R: GCTCTCTCCAAAAGCTCTTCC		
*β-actin*	F: CCTCTTCCAGCCATCTTTCTT	60	110
	R: TGTTGGCATACAGGTCCTTAC		

## Data Availability

The original contributions presented in the study are included in the article. Further inquiries can be directed to the corresponding author.

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
