# Peer review of "Changes in Ovarian Activity and Expressions of Follicle Development Regulation Factors During the Laying–Incubation Cycle in Magang Geese"

_animals, 2025, doi:10.3390/ani15101390_

Round 1

Reviewer 1 Report

Comments and Suggestions for Authors

The authors checked the changes of ovarian activity and expressions of follicle development regulation factors during the laying-incubation cycle in geese. The findings are novel. The authors tried to establish some factors during laying-incubation cycle. Therefore it will be great if authors could study this paper (https://doi.org/10.1111/asj.13261) where hypothalamic genes are studied in chicken. This information could be used in discussion to compare and contrast what's going on other poultry species. 

Reviewer 2 Report

Comments and Suggestions for Authors

This study focuses on the reproductive hormones and gene expression patterns during the laying-incubation cycle of Magang geese, a brood-prone breed extensively reared in southern China. Although the research addresses an important topic, it is marred by several shortcomings in both the manuscript's presentation and the experimental design. Significant revisions are necessary to enhance the clarity, rigor, and overall quality of the work.

  1. The writing should be improved

It is necessary to clearly define abbreviations ( such as LWF, SYF, LYF ) and mark the full name when it appears for the first time.

Line 35, F3G, define abbreviations.

Gene names. For example, StAR, STAR.

Lines 89, 138, and 146: Please clarify the meaning of GSI (Gonadosomatic Index), describe its calculation method, and ensure consistent and accurate use of the abbreviation "GSI" in figures and text.

  1. Abstract

line 33-35. Line 33 Make it clear in which follicle FSHR and GnIH expression  down regulated?  

  1. Introduction

I think the authors should introduce need and meaning for study of laying-incubatoin cycles, rather than introduce so much GnIH. These could be placed in the discussion parts for expresiong the results.

The overall literature reference is mostly the literature of distant years, and it is recommended to use the literature reference in recent years.

It is suggested that ‘might’ in ‘GnIH/GnIHR might play an important regulatory role’ should be adjusted to ‘suggest’ or ‘indicate’.

  1. Materials and methods

Animals Experimental Design parts. It is well known that Magang geese undergo several laying-incubation cycles during a full reproductive year. The authors should clarify whether the cycles used in this study were at the beginning, middle, or end of the reproductive period. Additionally, since the light schedule is a critical factor influencing reproductive behavior, the specific light regimen employed in the study should also be clearly stated. Also, the specific operation of forcibly terminate the broodiness (such as isolation duration, environmental conditions) may affect hormone levels, and additional methodological details are needed.

line 97-99, catalog number of the ELISA kit, as well as the detection limit and sensitivity of the controls, need to be specified.

Line 132. The Tukey test for multiple comparisons is not mentioned in the statistical methods. Additional explanation is needed to enhance the reliability of the results.

The applicability of ‘β-actin’ as an internal reference gene in RT-qPCR needs to be verified (such as citing stability analysis literature). Table 1, add beta actin primer.

  1. Results and discussion

The number of samples used in each figure should be provided, and please improve the clarity of Figure1, 2, 3,4,5,7.

Figure 1, add the abbreviations of each stage below the full names, as the descriptions of each stage in the results section use abbreviations. Including abbreviations in the image can facilitate easier reading.

WB figures for LHR and FSHR appear blurry, and the authors need to update the images. Additionally, why did the authors use β-actin as the internal reference in figure 5 and GAPDH in figure7?

For Figures 6 and 7, at which stage were the samples collected? It appears that the differences in gene expression across various follicular development stages are not directly related to the laying-incubation cycle. To fully address the study's objectives, it would be beneficial to investigate the gene expression of GnIH and GnRH in the hypothalamus, as well as FSH, FSHR, PRL, GnIHR, and GnRHR in the pituitary during the laying-incubation cycle. Additionally, examining gene expression in the LW ans SW which exist in all the reproduction stages could be meaningful.

Some of the gene expression changes ( such as GnIH mRNA and protein levels ) are not completely consistent, and the potential reasons need to be explained in the discussion.

Line 295-297. Lines 295-297: This conclusion is too assertive and cannot be drawn from the existing literature.

The conclusion of the article refers to ‘unpublished data’, and specific experimental results need to be deleted or supplemented to support the conclusion.

Line 231: please, check the sentence.

please, the core regulatory role of GnIH/GnIHR is summarized, and the research limitations and future development prospects are pointed out.

Comments on the Quality of English Language

must be improved

Reviewer 3 Report

Comments and Suggestions for Authors

Manuscript No. animals-3491571
"Changes of ovarian activity and expressions of follicle development regulation factors during the laying-incubation cycle in Magang geese"
is devoted to the study of the mechanisms of hormonal regulation of the reproductive function in geese. These studies are of considerable interest to specialists engaged in breeding local breeds of geese, since geese, unlike other bird species, are distinguished by a pronounced seasonality of reproduction and some breeds have retained the instinct of brooding.
The authors conducted a large volume of research on a significant population using modern research methods. The results of the work are presented and analyzed in detail. However, I consider it necessary to make some amendments to the manuscript.
1. The names of animal species are written with a capital letter and in italics. For example, line 47, reference â„–12, 18, etc.
2. lines 87-88 "on the 2nd day".
3. In the "Materials and Methods" section, information about the light regime should be added, since it is the duration and brightness of lighting that has a huge impact on hormone levels and reproductive function in general.
Add information about the number of samples used to assess ovarian morphology, for blood tests, etc. There is no data on the number of samples in any of the figures.
4. In the "Discussion" section, more attention should be paid to discussing the results of your own research. In fact, out of 92 lines of "Discussion", only 10 lines are devoted to your own research. It would be desirable to indicate the possibility of practical application of the research results.
5. Authors should carefully check the list of references and correct the spelling of the names of hormones and proteins - in capital letters (No. 3,5,6,7, 13, 17, 20......)
I propose to accept this article for publication after making the indicated corrections.

Reviewer 4 Report

Comments and Suggestions for Authors

The authors provided interesting data about the changes of ovarian activity and expressions of follicle development regulation factors during the laying-incubation cycle in Magang geese.

The design of this experiment is appropriate but the authors have not shied away from some shortcomings here, and the description of the results is accurate. The literature used in the manuscript is appropriate (References section verified: each item is in the available databases) . However, there are still some shortcomings in the current study and the authors should modify this manuscript according to the following suggestions.

  1. In the introduction, it would be appropriate to add information on why the reproductive aspect of the Magang Goose is important from the breeder's point of view (economic factors)
  2. The images added as an attachment are very blurry and merge with the background, making interpetation difficult. They should be corrected.
  3. The biggest deficiency is in the Materials and Methods section:

- To assess ovarian morphology it was the birds that had to be killed or sterilized.

- Under what environmental and nutritional conditions the birds were kept

- Was there approval from the Ethics Committee for this type of study?

- undetailed statistical description - Have the normal distribution and homogeneity of variance been checked? To apply these tests must be performed. This information should be completed here. Did all the data meet the above conditions. If not, what non-parametric tests were used?
